# Even Dispersion Design for a Compact Linear Loudspeaker Array with Adaptive Genetic Algorithm

**Juanjuan Cai [1], Yongqiang Pang [2], Hui Wang [1,*] and Yutian Wang [1]**

[1]   Key Laboratory of Media Audio & Video (Communication University of China), Ministry of Education, Beijing 100024, China; caijuanjuan@cuc.edu.cn (J.C.); wangyutian@cuc.edu.cn (Y.W.)

[2]   School of Information and Communication Engineering, Communication University of China, Beijing 100024, China; yq.pang@cuc.edu.cn

*   Correspondence: hwang@cuc.edu.cn

**Abstract:** Even dispersion is important for live sound reinforcement systems; however, it needs to be carefully designed when using a linear loudspeaker array. This is because the audience area is often large, while the loudspeakers are placed centrally in this case for convenience, and thus both the level and the frequency balance may not remain reasonably constant for all audiences. To solve this problem, the adaptive genetic algorithm is firstly introduced in the parameters optimization. Secondly, taking the radiation characteristics at different frequencies into account, a linear-phase non-uniform filter bank is proposed to decompose the broad frequency band into several sub-bands. The audio is re-synthesized with the optimized parameters in each frequency band for a linear loudspeaker array. To show the validity of the proposed method, the simulations and the experiments are conducted to demonstrate that the sound pressure level in the audience area is distributed within $\pm 1.33$ dB, ranging from 200 Hz to 4000 Hz.

**Keywords:** linear loudspeaker array; constant sound pressure level; broadband

## 1. Introduction

Live sound reinforcement aims to maintain constant listening experience for a large amount of audiences in real listening environments such as theater, cinema, auditoria, and stadium. The idea of the "democracy of sound" is a recent theme within the live sound reinforcement community [1]. Audiences should receive constant content regardless of the seat location. That is to say, for all audiences, the level and the frequency balance should stay nearly constant in the area that the loudspeakers are designed to cover. This paper studies the use of a compact linear loudspeaker array to achieve the even dispersion goal for practical applications.

Sound pressure level (SPL) distribution is one of the most popular objective measurements in studying the even/uneven dispersion of sound fields. To obtain a constant SPL in the audience area, line loudspeaker arrays have been well designed for sound reinforcement by many researchers. There are mainly two alternative groups of approaches in this field [2]. One is to compute the loudspeaker driving signals with a reproduction equation such as Ambisonics [3,4], the least squares (LS)-based solution [5,6], delay-and-sum beamformer [7,8], and so on. The excitation parameters of loudspeakers are calculated for matching the desired and the reproduced SPL distribution with the control points and the loudspeaker locations. However, these approaches are quite unstable because of the ill-conditioned problem [9] and sufficient only for the specific sound field with high accuracy [10]. The other is the well-known wave field synthesis (WFS) [11], which is derived from the Kirchhoff–Helmholtz integral [12]. For the planar and the linear arrays, the formulation is equivalent to Rayleigh's first integral formula [13]. However, it is a quite complicated system with a large number of loudspeakers which must be set up carefully and driven individually [14,15]. In this paper, an adaptive genetic

algorithm (GA)-based approach is proposed to compute the excitation parameters of loudspeakers. GA, firstly proposed by Holland [16], is an adaptive and competent approach for evolutionary search. It has been successfully applied in many fields; array optimization is an example [17–21], and can be treated as a kind of symbolic regression [22,23].

However, for real audio applications, it is necessary to consider the performance in a wide frequency range. Broadband designs are addressed in sound reproduction [24,25] and beamforming techniques [26]. There are some inevitable limitations for broadband sound field control with a uniformly spaced array [27]. With the increase of frequency, difficulties increase dramatically for constant sound radiation characteristics design. Firstly, the number of loudspeakers should be increased with the range of the frequency band. An array, which consists of 300 loudspeakers, is proposed by Nasim [25] for the signals up to 4000 Hz. However, only a limited number of loudspeakers is available for many practical applications. Lasso-based methods [6,27–29] are approved for loudspeaker location optimization with a reduced set of loudspeakers. Due to the complicated computation mentioned above, it is valuable to exploit a feasible system in real applications.

In this paper, we focus on providing the broadband constant SPL distribution in the audience area using a linear loudspeaker array ranging from 200 Hz to 4000 Hz. Firstly, an adaptive GA, which can overcome both the low speed and the premature convergence problems [30], is introduced for the even dispersion goal in narrowband. The proposed method shows good performance compared with conventional approaches such as delay and sum beamformer [7,8], contrast maximization [31], and LS-based pressure matching [32,33]. Secondly, non-uniform filter banks (NUFBs) [34,35], which have a linear-phase (LP) property, are proposed to divide the whole frequency band into sub-bands so that the optimized excitation parameters can be applied to each separate sub-band without causing phase variation. The results demonstrate that SPL distribution can be controlled within $\pm 1.33$ dB in the broadband.

The following parts of the paper are organized as follows. In Section 2, the excitation parameters optimization is introduced for the even dispersion goal in the narrowband case. For the broadband, the method is introduced in Section 3. Then, the simulation and the experiment results are, respectively, shown in Sections 4 and 5. Finally, the conclusions are drawn in Section 6.

**Notation**

Throughout this paper, we use the following notations: Matrices and vectors are represented by upper- and lowercase boldface, respectively, e.g., $\boldsymbol{X}$ and $\boldsymbol{x}$. $|\boldsymbol{x}|$ denotes the Euclidean distance where $|\boldsymbol{x}| = \sqrt{\boldsymbol{x} \cdot \boldsymbol{x}}$. $\boldsymbol{X}^{-1}$ and $\boldsymbol{X}^{+}$ denotes inverse and the generalized inverse of matrix $\boldsymbol{X}$. The denary logarithm is denoted by $\log_{10}(\cdot)$. $\max(\cdot)$ denotes the max function. $\exp(\cdot)$ denotes the Euler's number-based exponential function. $G^{H}$ is the Hamiltonian matrix of $G$. $\text{eign}(\cdot)$ denotes the eigenvector of the computing matrix. $\sin(\cdot)$ is the sinusoidal function. $\cos(\cdot)$ denotes the cosine function.

## 2. Excitation Parameters Optimization in Narrowband Case

To maintain constant SPL for audiences in a line for all seats, this section introduces the methods on deriving the excitation parameters in an offline manner (the proposed optimization algorithm can be extended for online estimation of the excitation parameters when a microphone array is placed at the line for all seats, which will be studied in future) for a line loudspeaker array. The placement of the loudspeaker array is plotted in Figure 1. The aim is to generate constant SPL for audiences sitting in line $AB$ by using this compact loudspeaker array whose center is the point $O$. The angle between $OA$ and $OB$ is $45°$, which means $AB = OB$. It is assumed that the sound field propagates under free-field conditions.

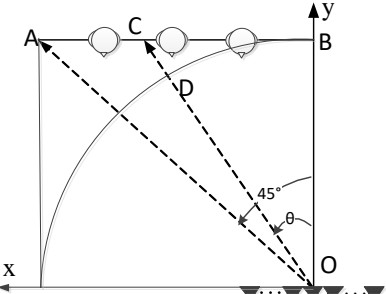

**Figure 1.** The displacement of the compact line loudspeaker array and the seats of audiences.

### 2.1. Theoretical Basis

The SPL at an observation position $x = [x, y]$ in a line is given as

$$P(x, k) = \int_{-\frac{l}{2}}^{\frac{l}{2}} D(x_0, k) G_{2D}(x, x_0, k)\, dx_0, \tag{1}$$

where $l$ is the length of the line array. $k$ is the wave number that relates to the frequency $f$. $x_0 = [x, 0]$ refers to one position of the line array. $D(x_0, k)$ denotes the sound driving function at this position, which can be given by

$$D(x_0, k) = A(x_0, k)\exp(-j\alpha(x_0, k)). \tag{2}$$

$G_{2D}(x, x_0, k)$ is the two dimensional free-field Green's function defined in [36]

$$G_{2D}(x, x_0, k) = \frac{\exp(-jk|x - x_0|)}{4\pi|x - x_0|}. \tag{3}$$

When the observation position is far away from the center of the line loudspeaker array with a fixed distance $r = |x - x_0|$, the sound pressure at angle $\theta$ can be reduced to [37]

$$\widetilde{P}(\theta, k) = \frac{1}{r} \sum_{n=1}^{N} D(x_{0,n}, k)\exp(-jknd_k\sin(\theta)), \tag{4}$$

where $N$ is the number of loudspeakers. $x_{0,n} = [x_n, 0]$ is the postion of the $n$th loudspeaker. $d_k$ is the interspace of thetwo adjacent loudspeakers which could be different in different frequencies. $\theta$ is illustrated in Figure 1. The parameters that need to be optimized are the magnitudes $A(x_0, k)$ and the phases $\alpha(x_0, k)$ in $D(x_0, k)$.

The directivity is defined as

$$R(\theta, k) = 20\log_{10}\frac{|\widetilde{P}(\theta, k)|}{\max(|\widetilde{P}(\theta, k)|)}. \tag{5}$$

### 2.2. Conventional Approaches

There are several approaches to compute the excitation parameters for the desired sound field.

#### 2.2.1. Delay and Sum Beamformer

Delay and sum beamformer is a simple method to control the sound field by setting control points. The sound driving function can be given by [7]

$$D_{DS}(x_0, k) = \sum_{m=1}^{M} \frac{1}{G_{2D}(x_m, x_0, k)}, \tag{6}$$

where $x_m$ is the control point and $x_0$ is the speaker position. $M$ is the number of control points.

### 2.2.2. Contrast Maximization

In contrast maximization approach, we can set control points at target and non-target regions. The spatial correlation between control points $x$ and loudspeaker locations $x_0$ is defined as [31]

$$C(x, x_0, k) = \frac{1}{M} \sum_{m=1}^{M} G_m^H(x, x_0, k) G_m(x, x_0, k), \tag{7}$$

where $G_m(x, x_0, k) = [G_{2D}(x_m, x_{0,1}, k)...G_{2D}(x_m, x_{0,n}, k)]$. The superscript $H$ denotes the Hamiltonian matrix operator.

The sound driving function can be given by

$$D_{CM}(x_0, k) = \text{eign}([C_{tar}(x, x_0, k) + C_{nontar}(x, x_0, k)]^+) C_{tar}(x, x_0, k), \tag{8}$$

where $C_{tar}(x, x_0, k)$ and $C_{nontar}(x, x_0, k)$ are the spatial correlation matrix between target control points and loudspeaker positions and that between non-target control points and loudspeaker positions, respectively.

### 2.2.3. LS-Based Pressure Matching

In LS-based pressure matching approach [38], the driving function is given by

$$D_{LSPM}(x_0, k) = G^{-1}(x, x_0, k) P(x, k), \tag{9}$$

where $G^{-1}(x, x_0, k)$ is the inverse of free-field Green's function between control points $x$ and loudspeaker locations $x_0$. Two sets of control points are designed with higer and lower values.

### 2.3. Proposed Approaches

To obtain constant SPL in line AB as shown in Figure 1, the generated SPL should compensate the attenuation in sound propagation because of the varying distance between the array and points in line AB. Since the SPL produced by a point sound source is defined as [39]

$$L = L_W - 20\log_{10}(r) - 11, \tag{10}$$

where $L_W$ is determined by the power of the source and $r$ is distance from the source to the measured point. If the SPL at point A is chosen as the reference, the SPL at the point C in Figure 1 can be given by

$$R_{ref}(\theta, k) = R_{ref}(45°, k) + 20\log_{10}\frac{OC}{OA} = R_{ref}(45°, k) + 20\log_{10}\frac{1}{\sqrt{2}\cos(\theta)}. \tag{11}$$

To generate such sound field, an adaptive GA-based approach is proposed for array excitation parameters optimization. The GA is a search algorithm motivated by the Darwin's natural selection theories. Unlike the above mentioned traditional methods, GA can search parallelly for the global optimal solutions among the possible spaces such as the excitation parameters in this paper. Promoted by the crossover and the mutation operators, the individuals that satisfy a predefined goal will survive in the evolution of a population which is similar to the notions of natural evolution. The GA is chosen because the aim is to get the best parameters in the search spaces within a relatively short time. Moreover, it can be handled easily to maintain multiple targets: (1) The target SPL distribution in the audience area. (2) The SPL constraint in unconcerned area (sidelobe of the array pattern). It is a nonlinear problem when the sidelobe is required to be lower than a threshold which can not be solved by the above mentioned traditional methods generally. The sidelobe control is important for practical applications due to that the SPL distribution in sidelobe should be low enough to avoid interference

with others. Note should be given that, although the low speed convergence problem is not a big problem for offline excitation parameters optimization, the premature convergence problem could degrade the performance a lot for practical applications. As mentioned above, when extending the adaptive GA to online excitation parameters optimization, the adaptive GA is expected to solve all the two problems of the traditional GA.

To describe the GA procedure, it is necessary to define the chromosome and the fitness function. The chromosomes, also known as individuals, represent the possible solutions in the search spaces. For simplicity, only the phases $\alpha(x_0, k)$ are optimized in the following. The phases for loudspeaker array, range from $0°$ to $360°$, can be encoded and spliced into a binary string. Quantitative accuracy is a parameter for the chromosome binary encoding. For example, if the phases are $\{\alpha(x_{0,0}, k)\ \alpha(x_{0,1}, k) \ldots \alpha(x_{0,N-2}, k)\ \alpha(x_{0,N-1}, k)\}$, where $N$ is the number of loudspeakers, a chromosome for these parameters can be encoded as {01000 01010 ... 01011 11001}. New individuals can be decoded into phases for fitness calculation in the selection operator.

To obtain a more precise representation of the directivity pattern, the objective function is defined as [40]

$$O(\theta, k) = a_1 \sum_{0° < \theta < 45°} [|R(\theta, k)| - R_{ref}(\theta, k)]^{b_1} + a_2 [|\max_{-90° < \theta < 0°} (R(\theta, k))|]^{b_2}, \tag{12}$$

where $a_1$, $a_2$, $b_1$, and $b_2$ refer to the controlled coefficients. This function requests an optimized individual that should maintain (a) minimum differences between the objective distribution defined in Equation (12) and the optimized distribution defined in Equation (5) with decoded phases, (b) minimum SPL distribution in the sidelobe. Unlike the traditional methods, an exact objective sidelobe distribution is not well considered in the sidelobe control.

The fitness function is the transformation of the objective function [7,41]

$$J(\theta, k) = \frac{c}{O(\theta, k)} + d, \tag{13}$$

where $c$ and $d$ are controlled coefficients in the experiments.

The adaptive GA procedure is summarized in Algorithm 1. The initial populations are generated randomly. Each individual in the populations is awarded a fitness score computed by the fitness function after decoding its chromosome. After that, three basic genetic operators are conducted: Selection, crossover, and mutation. The selection operator is defined according to the fitness. To achieve diversity and convergence, the higher the fitness individual, the more likely to survive in the selection operator. In the crossover process, new generations are produced by the individuals after selection. A single-point crossover operator is applied in this paper to generate new individuals by dividing parent chromosomes into some parts and inheriting from different parents. The mutation operator adds diversity to the population and explores new parameter search spaces. The generations with optimized phases $\alpha(x_0, k)$ can be produced when some stopping criteria become true.

---

**Algorithm 1** Adaptive GA

---

　**Step 1.** Set population size $n_{pop}$ and the coefficients in the adaptive fitness function.
　**Step 2.** Generate initial population for the phases $\alpha(x_0, k)$ in the possible search space.
　**repeat**
　　**Step 3.** Calculate the fitness for each chromosome by Equation (13). Compute crossover rate and
　　mutaion rate.
　　**Step 4.** Peform selection based on the fitness.
　　**Step 5.** Peform crossover on the generation.
　　**Step 6.** Peform mutation on the generation.
　**until** Satisfying the stopping criteria with optimized $\alpha(x_0, k)$.

---

## 3. Constant SPL Distribution Design for Broadband Signals

As mentioned above, the constant SPL distribution design is implemented in a narrowband case. However, due to the different radiation characteristics of high and low frequencies, it needs great efforts to achieve this goal for broadband applications. In this section, according to the radiation characteristics of a loudspeaker array on different frequencies, we divide the broadband into many independent sub-bands to apply the optimized parameters separately.

### 3.1. Frequency Division

Because of the different radiation characteristics, the frequency band should be divided properly to get better performance in each sub-band. With an equispaced array, the radiation characteristic varies with the interspace of the adjacent loudspeakers, the number of loudspeakers, and the excitation frequency. We take the arrays with 4, 6, and 8 loudspeakers, respectively, as examples to describe the division strategy with a fixed interspace of the two adjacent loudspeakers.

In the specific sub-band, the same delay should be applied since the phase is a frequency dependent parameter. With the phase $\alpha(x_0, k_{ct})$ computed by adaptive GA at the central frequency of the same sub-band, the phase at the noncentral frequency is

$$\alpha(x_0, k') = \alpha(x_0, k_{ct}) \frac{k'}{k_{ct}}, \tag{14}$$

where $k'$ and $k_{ct}$ refer to the wave number at the noncentral and that at the central frequency, respectively. One can see that the phase at the band edge may be significantly influenced when the bandwidth is too large. The SPL in the target line can be calculated by

$$\widetilde{R}(x_0, k') = R(\langle x_C, x_B \rangle, k') + 20\log_{10}(\cos\langle x_C, x_B \rangle), \tag{15}$$

where $R(\langle x_C, x_B \rangle, k')$ is the directivity computed by Equation (5) with the optimized phase $\alpha(x_0, k')$. $x_C$ is the position at the point C as shown in Figure 1. $x_B$ denotes the position of point B.

To optimize the proper bandwidth, the rule for the bandwidth decision can be given by

$$B_{res} = \max_{|\widetilde{R}(x_0,k') - \bar{\widetilde{R}}(x_0,k')|<1.5dB} (k' - k_{ct})c/\pi, \tag{16}$$

where $c$ denotes the speed of sound. $\bar{\widetilde{R}}(x_0, k')$ is the average value of $\widetilde{R}(x_0, k')$ over $x_0$. The SPL distrubution $\widetilde{R}(x_0, k')$ should meet the predefined performance requirement with maximum bandwidth, where the SPL difference over the target line should be less than $\pm1.5$ dB [42] in each sub-band. Besides, the band division strategy should obey the feasible condition in the NUFBs design [34]: For the $l$th band, $\sum_{i=0}^{l-1} B_{res,i}$ equals to $B_{res,l} v_l$ with some integer $v_l$. Hence, there are only a finite number of bandwidth choices in each sub-band design which can be solved with the enumeration method.

As a result, for the line loudspeaker array with four loudspeakers, the relative SPL distributions of the lower and the upper frequencies in the first band are shown in Figure 2a,b, respectively. With Equation (16), we can obtain the bandwidth in each band. For the central frequency from 200 Hz to 600 Hz, the bandwidth is 100 Hz. While for 600 Hz–1200 Hz, the bandwidth is 200 Hz. When the central frequency ranges from 1200 Hz to 2000 Hz, the bandwidth is 400 Hz. The remainder frequency band is divided into three parts, which are 2000 Hz–2500 Hz, 2500 Hz–3000 Hz, and 3000 Hz–4000 Hz. The lower and the upper frequencies of each band for 4 loudspeakers are summarized in Table 1.

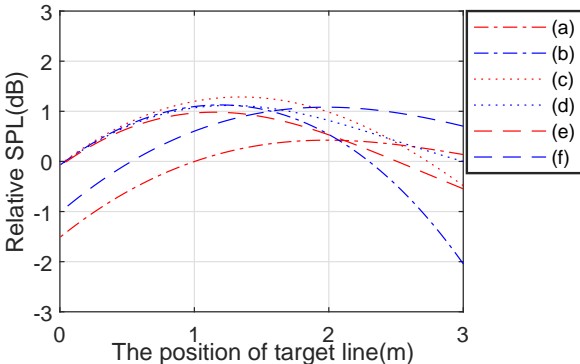

**Figure 2.** The radiation results comparison with four loudspeakers at (**a**) 200 Hz and (**b**) 300 Hz, six loudspeakers at (**c**) 200 Hz and (**d**) 250 Hz, and eight loudspeakers at (**e**) 200 Hz and (**f**) 220 Hz.

**Table 1.** The band and its corresponding lower feaquency $f_l$, upper frequency $f_u$ and bandwidth $B_{res}$ for the line loudspeaker array with $N = 4, 6, 8$ loudspeakers.

| N | Band | $f_l$ (Hz) | $f_u$ (Hz) | $B_{res}$ (Hz) | N | Band | $f_l$ (Hz) | $f_u$ (Hz) | $B_{res}$ (Hz) |
|---|---|---|---|---|---|---|---|---|---|
| 4 | 1 | 200 | 300 | 100 | 4 | 7 | 1000 | 1200 | 200 |
| 4 | 2 | 300 | 400 | 100 | 4 | 8 | 1200 | 1600 | 400 |
| 4 | 3 | 400 | 500 | 100 | 4 | 9 | 1600 | 2000 | 400 |
| 4 | 4 | 500 | 600 | 100 | 4 | 10 | 2000 | 2500 | 500 |
| 4 | 5 | 600 | 800 | 200 | 4 | 11 | 2500 | 3000 | 500 |
| 4 | 6 | 800 | 1000 | 200 | 4 | 12 | 3000 | 4000 | 1000 |
| 6 | 1 | 200 | 250 | 50 | 6 | 10 | 900 | 1000 | 100 |
| 6 | 2 | 250 | 300 | 50 | 6 | 11 | 1000 | 1200 | 200 |
| 6 | 3 | 300 | 350 | 50 | 6 | 12 | 1200 | 1400 | 200 |
| 6 | 4 | 350 | 400 | 50 | 6 | 13 | 1400 | 1600 | 200 |
| 6 | 5 | 400 | 500 | 100 | 6 | 14 | 1600 | 1800 | 200 |
| 6 | 6 | 500 | 600 | 100 | 6 | 15 | 1800 | 2000 | 200 |
| 6 | 7 | 600 | 700 | 100 | 6 | 16 | 2000 | 2500 | 500 |
| 6 | 8 | 700 | 800 | 100 | 6 | 17 | 2500 | 3000 | 500 |
| 6 | 9 | 800 | 900 | 100 | 6 | 18 | 3000 | 4000 | 1000 |
| 8 | 1 | 200 | 220 | 20 | 8 | 15 | 780 | 840 | 60 |
| 8 | 2 | 220 | 240 | 20 | 8 | 16 | 840 | 900 | 60 |
| 8 | 3 | 240 | 260 | 20 | 8 | 17 | 900 | 1000 | 100 |
| 8 | 4 | 260 | 280 | 20 | 8 | 18 | 1000 | 1100 | 100 |
| 8 | 5 | 280 | 320 | 40 | 8 | 19 | 1100 | 1200 | 100 |
| 8 | 6 | 320 | 360 | 40 | 8 | 20 | 1200 | 1350 | 150 |
| 8 | 7 | 360 | 400 | 40 | 8 | 21 | 1350 | 1500 | 150 |
| 8 | 8 | 400 | 450 | 50 | 8 | 22 | 1500 | 1650 | 150 |
| 8 | 9 | 450 | 500 | 50 | 8 | 23 | 1650 | 1800 | 150 |
| 8 | 10 | 500 | 550 | 50 | 8 | 24 | 1800 | 2100 | 300 |
| 8 | 11 | 550 | 600 | 50 | 8 | 25 | 2100 | 2400 | 300 |
| 8 | 12 | 600 | 660 | 60 | 8 | 26 | 2400 | 2800 | 400 |
| 8 | 13 | 660 | 720 | 60 | 8 | 27 | 2800 | 3200 | 400 |
| 8 | 14 | 720 | 780 | 60 | 8 | 28 | 3200 | 4000 | 800 |

For the line loudspeaker array with six and eight loudspeakers, the same frequency band division strategy can be applied with Equation (16). Specifically, for six loudspeakers, the relative SPL distributions of the lower and the upper frequencies in the first band are plotted in Figure 2c,d. For eight loudspeakers, the bandwidth of first sub-band is 30 Hz, where its relative SPL distributions of the lower and the upper frequencies are shown in Figure 2e,f. For the all the frequencies ranging from 200 Hz to 4000 Hz, the lower and the upper frequencies of each band are presented in Table 1 for six and eight loudspeakers.

One can see from Table 1 that, with the increase of the number of loudspeakers, the frequency band should become narrower. With a fixed number of loudspeakers, the frequency band becomes wider at high frequencies.

At high frequencies, the interspace of the array should be smaller due to shorter wavelengths than those of low frequencies. For the following experiment and analysis, the used loudspeaker array are shown in Figure 3. The sub-array is named as "Group 1" with the interspace 20 cm for 200 Hz to 1000 Hz. The sub-array "Group 2" with the interspace 10 cm is used for the band 1000 Hz–02000 Hz. For higher frequencies (>2000 Hz), the sub-array "Group 3" with the interspace 5 cm is chosen.

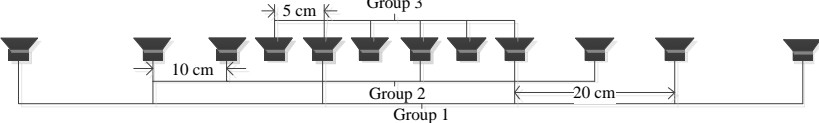

**Figure 3.** The designed loudspeaker array.

### 3.2. Digital Filter Bank Design

To handle the different radiation characteristics, a digital filter bank is chosen to perform the frequency division. It is necessary to achieve LP property for accurate phases in each sub-band. The design of LP-NUFBs [35] with the efficient partial modulation technique can generalize the phase modification structure for the LP property. Compared with traditional NUFB designs, it has much lower system delay and computational complexity when using LP-NUFBs, while a promising performance can be achieved.

The band division result should be transformed into the angular frequency domain. The corresponding angular frequency $\omega$ can be calculated as

$$\omega = \frac{2\pi f}{f_s}, \tag{17}$$

where $f$ ranging from 200 Hz to 4000 Hz and $f_s$ is the sampling rate.

To ensure the frequency response performance in the narrow frequency band of the filter bank, the order of the filter bank is set to 1500. In the band division result of 6 loudspeakers, the sampling factors are $[1/40, /160, 1/160, 1/160, 1/160, 1/80, 1/80, 1/80, 1/80, 1/80, 1/80, 1/40, 1/40, 1/40, 1/40, 1/40, 1/16, 1/16, 1/8, 1/2]$. The designed 20-channel LP-NUFBs is shown in Figure 4. The magnitude responses of the analysis filters are plotted in Figure 4a. The aliasing and the amplitude distortions are $E_a = 0.013$ and $E_{pp} = 0.041$, respectively, as shown in Figure 4b,c.

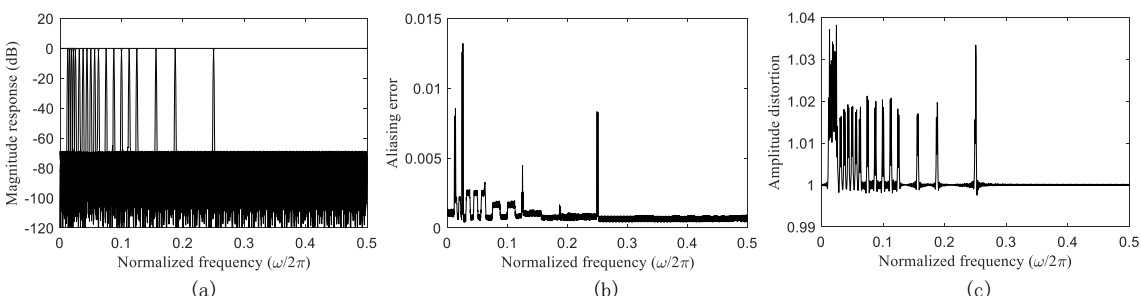

**Figure 4.** The designed 20-channel linear-phase non-uniform filter banks (LP-NUFBs). (**a**) Magnitude responses of the analysis filters. (**b**) Aliasing error. (**c**) Amplitude distortion.

### 3.3. Magnitude Response Compensation

The magnitude response is one of the most important factors for the perceived sound quality of loudspeaker [43]. A flat magnitude response [44,45] is desired to reproduce the input signals. In this work, we focus on the constant listening experience in the target area. It is also necessary to realize a flat magnitude response system for constant SPL distribution in the whole designed frequency band.

The measured magnitude response curve of a loudspeaker used in the experiment is shown in Figure 5a. The magnitude optimization goal is to compensate the non-flat magnitude response by using an inverse of the magnitude response. The compensating magnitudes are shown in Figure 5b.

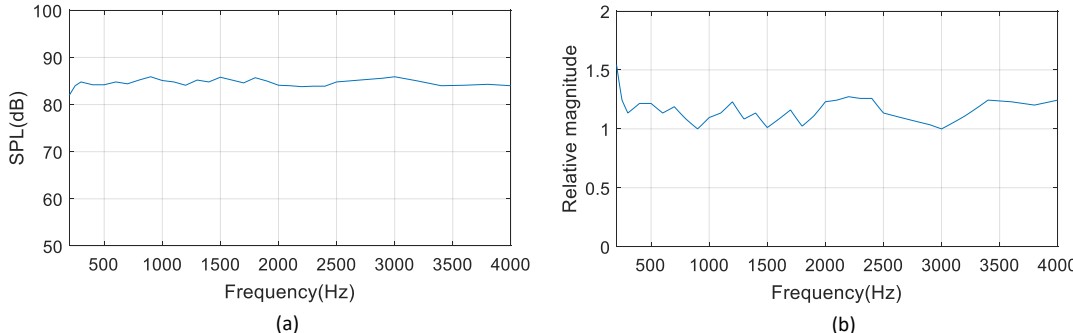

**Figure 5.** (**a**) The magnitude response curve. (**b**) The compensation magnitude curve.

The same calibration procedure can be conducted on each loudspeaker to achieve the same flat magnitude response. Note that this paper only considers the case that all the loudspeakers are the same type. If a line loudspeaker array consists of different types of loudspeakers, the optimization problem becomes more difficult, which can be a future research work. In the simulations, we assume that all the loudspeakers have the same flat magnitude response. In the experiments, we use the line loudspeaker array that all the loudspeakers are the same type and their magnitude responses have been compensated as describe in this part.

## 4. Simulation

With the LP-NUFBs, the whole frequency band can be divided into several sub-bands. Thus, the excitation parameters can be optimized separately in each sub-band by the proposed adaptive GA. In this section, we discuss the performance at some specific frequencies firstly. After that, a sweep signal with its frequency ranging from 200 Hz to 4000 Hz is applied to evaluate the performance for broadband signals.

### 4.1. Performance Evaluation of Narrowband Signals

The control points are located in a line with $y = 3$ m and $-3$ m$< x < 3$ m. With the delay and sum beamformer and the contrast maximization methods, sixty control points ($M = 60$) are discretized with $\Delta x = 0.1$ m. The LS-based pressure matching method is applied with six control points in the line with $y = 3$ m, $-3$ m$< x < 3$ m. The frequency is chosen to be 600 Hz.

Figure 6 shows the produced relative SPL in the target line with the proposed adaptive GA and the other conventional methods including delay and sum beamformer, contrast maximization, and LS-based pressure matching. Compared with the other two traditional methods, the LS-based pressure matching shows better performance for $x <= 2$ m where the SPL difference is within $\pm 9.9$ dB while it reduces rapidly for $x > 2$ m. The SPL distribution is nearly constant over the whole target line with the proposed adaptive GA. In all, the proposed adaptive GA with the initial population $n_{pop} = 200$ shows the best performance, where the SPL difference is within $\pm 0.72$ dB which is much better than the competing methods.

The performances of 500 Hz, 1000 Hz, 2000 Hz, and 4000 Hz with the proposed adaptive GA are shown in Figure 7. One can see that the SPL difference is less than 1.5 dB over the whole target line for all frequencies, which satisfies the requirements.

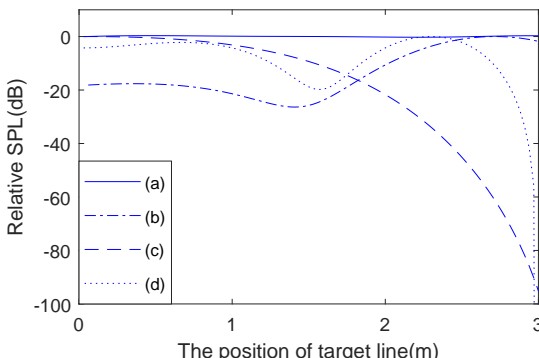

**Figure 6.** The relative sound pressure level (SPL) distribution in the target line with (**a**) adaptive genetic algorithm (GA), (**b**) delay and sum beamformer, (**c**) contrast maximization, and (**d**) least squares (LS)-based pressure matching approaches.

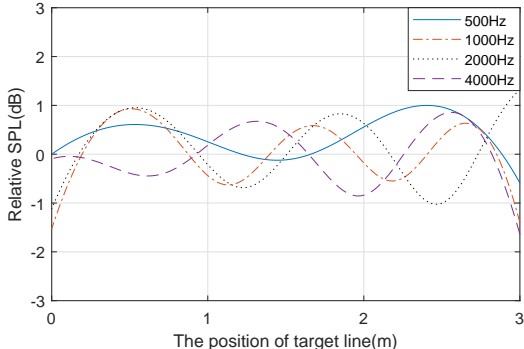

**Figure 7.** The relative SPL distribution of 500 Hz, 1000 Hz, 2000 Hz, and 4000 Hz.

### 4.2. Performance Evaluation of Broadband Signals

As described in Section 3, we divide the frequency pattern of the loudspeaker array with 12 loudspeakers. With the 20-channel LP-NUFBs design, the parameters calculated by the adaptive GA in each sub-band are applied in the whole frequency band.

To simulate the constant performance, a 20 s double swept-frequency signal is applied with the superposition of 200 Hz–2000 Hz and 400 Hz–4000 Hz swept-frequency signals which means there are that two frequency components $k_1$ and $k_2$ are always presented at the same time. Given that there are two separate sound sources with $\widetilde{P}(\theta, k_1)$ and $\widetilde{P}(\theta, k_2)$ radiated by the same loudpeaker array, the synthetic relative SPL can be given by [46]

$$\tilde{R}(\theta, k_1, k_2) = 20\log_{10}\left( \frac{\widetilde{P}(\theta, k_1) + \widetilde{P}(\theta, k_2)}{\max\limits_{\theta}\left(\widetilde{P}(\theta, k_1) + \widetilde{P}(\theta, k_2)\right)} \right). \tag{18}$$

At a specific time, the signal has two narrowband frequency components. Their sound pressures can be calculated separately with the phases optimized by adaptive GA. After that, the relative SPL at this specific time can be calculated with Equation (18). The SPL reproduced by the testing signals is shown in Figure 8a. It can be seen that the SPL in the target line tends to be consistent in the designed frequency range.

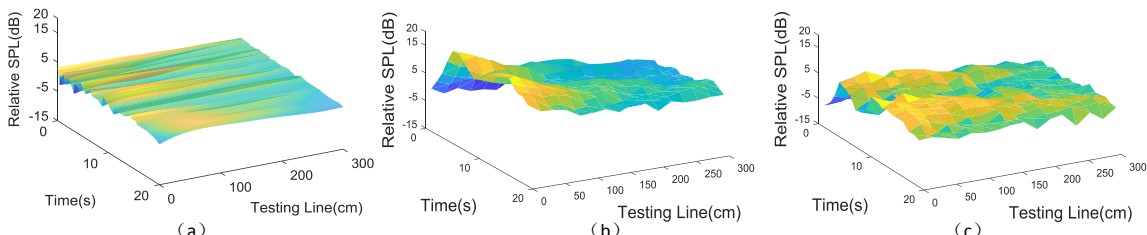

**Figure 8.** The results of simulation and experiments. (**a**) Simulated distribution of relative SPL. (**b**) Non-optimized time-varying distribution of SPL. (**c**) Optimized time-varying distribution of SPL.

## 5. Experiment

As described above, the optimized excitation parameters of the loudspeaker array can be calculated by adaptive GA. Next, experiments are conducted to verify the performance of our system design. In these experiments, the wave files are synthesized by the LP-NUFBs. The SPL in the target line is measured when playing the multi-channel sound signals with the loudspeaker array.

### 5.1. Experiment Preparation

In the experiments, the same double swept-frequency signal in the simulation is applied. To validate the effectiveness of the proposed method, the SPL distributions are measured with the non-optimized and the optimized audio signals. In the non-optimized experiment, the same testing audio signals are played by the loudspeaker array which means the delays and the amplitudes are the same for all the loudspeakers. For the optimized SPL distribution measurements, the excitation parameters including the phases optimized by adaptive GA and the optimized magnitudes for flat magnitude responses are applied by LP-NUFBs for multi-channel audios synthesis. The input signal is divided into several sub-bands by the LP-NUFBs in each frame. After applying the optimized excitation parameters in each sub-band, the multi-channel audio signals can be obtained separately for the loudspeakers. Note that, the overlap-save method [47] is conducted in the frame processing procedure to avoid time aliasing. The overlapped portions in frames are discarded dicrectly which makes it more efficient than the well-known overlap-add method [48]. The frame length is set to one second. The frameshift is 0.7 s. The sampling rate of the wave file is 48 kHz.

The experiment is conducted in the Key Laboratory of Media Audio & Video (KLMAV) Lab, Communication University of China. In these experiments, the line loudspeaker array is placed as shown in Figure 3. The multi-channel signals generating with optimized excitation parameters are played by the loudspeakers. The SPL distribution is measured by sound level meters TES-1353 in the target line , which is about 3 m away from the array. Each test point is spaced 15 cm.

### 5.2. Experimental Results and Analysis

Before conducting the experiments, the background noise of the target line is measured firstly to figure out the noise distribution. By doing so, the SPL values contributed by the loudspeaker array can be obtained by subtracting the noise SPL values to improve the evaluation accuracy.

#### 5.2.1. Non-Optimized SPL Distribution

The non-optimized experiment is conducted to prove the effectiveness of optimized results. The double swept-frequency signals with same phases and magnitudes are played directly by the loudspeaker array. The time-varying SPL distribution is measured in 20 s at an interval of 15 cm.

The SPL distribution for the non-optimized case is shown in Figure 8b, which indicates that the SPL distribution is much higher in the center of the loudspeaker array and it gradually reduces from the center to the left/right side. With the increase of the frequency, the mainlobe of the line loudspeaker

array tends to become narrower. One can also observe that the SPL distribution can range from $-5$ dB to 20 dB at high frequency.

### 5.2.2. Optimized SPL Distribution

With the same experiment setup, the proposed method is applied to optimize the double swept-frequency signal for all the loudspeakers. The time-varying SPL distribution of the proposed method shows much better performance than the non-optimized result as shown in Figure 8c. To show the validity of optimization directly, the SPL distributions in different positions within different times are normalized and averaged. In Figure 9, the non-optimized SPL distributions range in $\pm5.27$ dB. With optimization, the SPL distributions are much smaller, which range in $\pm1.33$ dB.

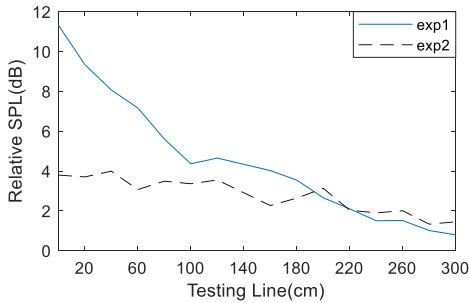

**Figure 9.** Comparison diagram of relative SPL distribution. "exp1" shows the non-optimized relative SPL distribution. "exp2" shows the optimized relative SPL distribution.

### 6. Conclusions

In this work, we present an efficient and effective method to achieve the even dispersion goal using a compact linear loudspeaker array. To obtain a constant SPL in the audience area, an adaptive GA is introduced for parameters optimization to get much better results compared with conventional approaches in the narrowband. For the real audio application, the broadband is divided into several narrowbands when introducing the linear-phase non-uniform filter banks before parameters optimization. The simulated and the experimental results validate that the SPL distribution is controlled within $\pm1.33$ dB. Future work should concentrate on optimizing parameters of multiple linear loudspeaker arrays for more complicated live audio reinforcement systems.

**Author Contributions:** Conceptualization, J.C. and H.W.; Data curation, J.C. and Y.P.; Formal analysis, H.W. and Y.W.; Investigation, J.C. and Y.P.; Methodology, J.C. and Y.P.; Validation, J.C., Y.P. and Y.W.; Writing-original draft preparation, Y.P.; Writing-review & editing, Y.P. and J.C.; Supervision, H.W. All authors have read and agreed to the published version of the manuscript.

**Acknowledgments:** This research is supported by the National Natural Science Foundation of China (Grant No. 61631016, 61231015), the National Radio and Television Administration Project (Grant No. 2015-53) and the Fundamental Research Funds for the Central Universities (Grant No. 2014XNG1425). The authors would like to thank the anonymous reviewers and the associate editor for their valuable comments that helped to improve the quality of this work. The authors would like to thank Prof. Hao and Zheng in Institute of Acoustics, Chinese Academy of Sciences in discussing and proofreading this manuscript.

**Conflicts of Interest:** The authors declare no conflict of interest.

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
