# Peer review of "Even Dispersion Design for a Compact Linear Loudspeaker Array with Adaptive Genetic Algorithm"

_applsci, doi:10.3390/app10010227_

Round 1

Reviewer 1 Report

report on the manuscript

Even dispersion Design for A Compact Linear
Loudspeaker Array with Adaptive Genetic Algorithm

submitted to applied science
by
Juanjuan Cai, Yongqiang Pang, Hui Wang, and Yutian Wang

The manuscript treats the question how to achieve an equally high
sound pressure level emitted by a loudspeaker line array, for a typically seated
audience. The authors try to achieve the equi-level for all frequencies by using a genetic algorithm for parameter optimization.
I might not have understood completely the approach, but it seems to me that there is no feedback loop involved, i.e. there are no microphones to feed back the truly measured sound at the positions to be optimized. This is a clear weakness of the paper, in any real situation it is not sufficient to only measure, but to control the signal (maybe by a GA, cf. https://arxiv.org/abs/1912.01412, https://openresearchsoftware.metajnl.com/articles/10.5334/jors.192/)

The main question that remains open for me is what is the advantage to use a genetic algorithm in contrast to generalized linear regression which is much faster. Given that the authors are hooked on performance this seems to me the first question to check. I might misunderstand again, but as far as I am concerned, if we read Up to equation 16, there is only analytical work involved and the equations are basically linear. One can easily use Levenberg-Marquardt (old, but works) to find optimal parameters, efficient, fast and in parallel.

The paper is well written, contains an answer to a relevant scientific question and the methods and results are well described. I doubt, though, that the method is optimal in terms of performance (even though I am a big fan of GA and GP). The originality of the work is not very expressed, this is a reasonable study of using an optimization method for the equilibration of SPL at a point.
Compared to other studies this is not a breaktrhough, but adds to the field.
I do not think that the present study is very significant, but it is significant enough to be published. I do not know a study of exactly this type from the literature. There are many similar studies, though,( e.g. https://doi.org/10.1121/1.1880852 and many more if you go for an internet search in the acoustic community). I would review the paper again after the authors have revised the paper with respect to
- language, there are typoos and missing articles etc., see some examples below
- the discussion of a comparison beyond LS of the GA method. This requires a bit of literature reading and comparing numbers *quantitatively*!
- a slightly deeper explanation of terms used, I fancy that non-experts will have a hard time to assign a chromosome with acoustics (if you know both worlds it makes perfect sense)

some smaller things to note:
- on p.1 l.28 conditioning -> conditioned
- In Eq. 7: what is the superscript H?
- p. 4 l 117: misprint controlleded

Author Response

Dear reviewer #1 and editor,

Please download the attachment to check the response.

Thank you for your valuable comments which helped us a lot to significantly improve the quality of this work.

Best wishes,

Yongqiang Pang

Reviewer 2 Report

This paper gives a competent treatment of a problem in loudspeaker array design. The authors have set themselves a relatively easy task, to equalise the sound intensity for listeners along a relatively short straight line. But this case allows them to give a clear comparison of their proposed procedure with various previous approaches, and the results are clear and convincing. I have only minor comments.

The only technical point: Figure 6a is described as the response for THE loudspeaker used in the experiments. But an array of speakers is used: how similar are they, in fact? Should a robust calculation procedure take some account of variability in the loudspeaker responses? This issue should at least be mentioned in the discussion.

Apart from this, the only issue is with the English language. The language is quirky but in most places it is quite understandable. However, some passages are very cryptic. For example, the description and caption of Figure 2 made me stop and think carefully to understand what is plotted. A more full caption would have helped a lot. But what does “right side frequency” mean, I wonder? The paper would benefit a lot from being read through by a native speaker of English.

Author Response

Dear reviewer #2 and editor,

Please download the attachment to check the response.

Thank you for your valuable comments which helped us a lot to significantly improve the quality of this work.

Best wishes,

Yongqiang Pang

Round 2

Reviewer 1 Report

Dear authors,

thanks for considering my comments, I have no further objections against publications. For any practical purpose I would consider a control loop with microphone on the audiences seats.